# Learning Hierarchical Information Flow
# with Recurrent Neural Modules

**Danijar Hafner** *
Google Brain
mail@danijar.com

**Alex Irpan**
Google Brain
alexirpan@google.com

**James Davidson**
Google Brain
jcdavidson@google.com

**Nicolas Heess**
Google DeepMind
heess@google.com

## Abstract

We propose ThalNet, a deep learning model inspired by neocortical communication
via the thalamus. Our model consists of recurrent neural modules that send features
through a routing center, endowing the modules with the flexibility to share features
over multiple time steps. We show that our model learns to route information
hierarchically, processing input data by a chain of modules. We observe common
architectures, such as feed forward neural networks and skip connections, emerging
as special cases of our architecture, while novel connectivity patterns are learned
for the text8 compression task. Our model outperforms standard recurrent neural
networks on several sequential benchmarks.

## 1 Introduction

Deep learning models make use of modular building blocks such as fully connected layers, convolu-
tional layers, and recurrent layers. Researchers often combine them in strictly layered or task-specific
ways. Instead of prescribing this connectivity a priori, our method learns how to route information as
part of learning to solve the task. We achieve this using recurrent modules that communicate via a
routing center that is inspired by the thalamus.

Warren McCulloch and Walter Pitts invented the perceptron in 1943 as the first mathematical model
of neural information processing [22], laying the groundwork for modern research on artificial neural
networks. Since then, researchers have continued looking for inspiration from neuroscience to identify
new deep learning architectures [11, 13, 16, 31].

While some of these efforts have been directed at learning biologically plausible mechanisms in an
attempt to explain brain behavior, our interest is to achieve a flexible learning model. In the neocortex,
communication between areas can be broadly classified into two pathways: Direct communication
and communication via the thalamus [28]. In our model, we borrow this latter notion of a centralized
routing system to connect specializing neural modules.

In our experiments, the presented model learns to form connection patterns that process input
hierarchically, including skip connections as known from ResNet [12], Highway networks [29], and
DenseNet [14] and feedback connections, which are known to both play an important role in the
neocortex and improve deep learning [7, 20]. The learned connectivity structure is adapted to the
task, allowing the model to trade-off computational width and depth. In this paper, we study these
properties with the goal of building an understanding of the interactions between recurrent neural
modules.

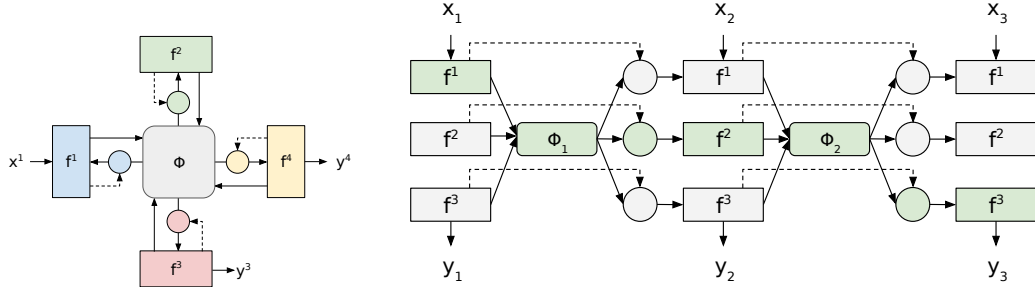

(a) Module $f^1$ receives the task input, $f^2$ can be used for side computation, $f^3$ is trained on an auxiliary task, and $f^4$ produces the output for the main task.

(b) Computation of 3 modules unrolled in time. One possible path of hierarchical information flow is highlighted in green. We show that our model learns hierarchical information flow, skip connections and feedback connections in Section 4.

Figure 1: Several modules share their learned features via a routing center. Dashed lines are used for dynamic reading only. We define both static and dynamic reading mechanisms in Section 2.2.

Section 2 defines our computational model. We point out two critical design axes, which we explore experimentally in the supplementary material. In Section 3 we compare the performance of our model on three sequential tasks, and show that it consistently outperforms multi-layer recurrent networks. In Section 4, we apply the best performing design to a language modeling task, where we observe that the model automatically learns hierarchical connectivity patterns.

## 2 Thalamus Gated Recurrent Modules

We find inspiration for our work in the neurological structure of the neocortex. Areas of the neocortex communicate via two principal pathways: The *cortico-cortico-pathway* comprises direct connections between nuclei, and the *cortico-thalamo-cortico* comprises connections relayed via the thalamus. Inspired by this second pathway, we develop a sequential deep learning model in which modules communicate via a routing center. We name the proposed model ThalNet.

### 2.1 Model Definition

Our system comprises a tuple of computation modules $F = (f^1, \cdots, f^I)$ that route their respective features into a shared center vector $\Phi$. An example instance of our ThalNet model is shown in Figure 1a. At every time step $t$, each module $f^i$ reads from the center vector via a context input $c_t^i$ and an optional task input $x_t^i$. The features $\phi_t^i = f^i(c_t^i, x_t^i)$ that each module produces are directed into the center $\Phi$.[2] Output modules additionally produce task output from their feature vector as a function $o^i(\phi^i) = y^i$.

All modules send their features to the routing center, where they are merged to a single feature vector $\Phi_t = m(\phi_t^1, \cdots, \phi_t^I)$. In our experiments, we simply implement $m$ as the concatenation of all $\phi^i$. At the next time step, the center vector $\Phi_t$ is then read selectively by each module using a reading mechanism to obtain the context input $c_{t+1}^i = r^i(\Phi_t, \phi_t^i)$.[3] This reading mechanism allows modules to read individual features, allowing for complex and selective reuse of information between modules. The initial center vector $\Phi_0$ is the zero vector.

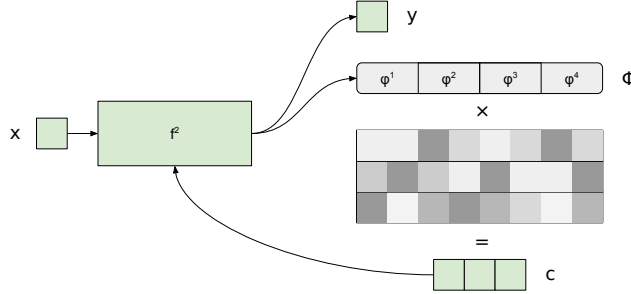

Figure 2: The ThalNet model from the perspective of a single module. In this example, the module receives input $x^i$ and produces features to the center $\Phi$ and output $y^i$. Its context input $c^i$ is determined as a linear mapping of the center features from the previous time step. In practice, we apply weight normalization to encourage interpretable weight matrices (analyzed in Section 4).

In summary, ThalNet is governed by the following equations:

Module features:
$$\phi_t^i = f^i(c_t^i, x_t^i) \tag{1}$$

Module output:
$$y_t^i = o^i(\phi_t^i) \tag{2}$$

Center features:
$$\Phi_t = m(\phi_t^1, \cdots, \phi_t^I) \tag{3}$$

Read context input:
$$c_{t+1}^i = r^i(\Phi_t, \phi_t^i) \tag{4}$$

The choice of input and output modules depends on the task at hand. In a simple scenario (e.g., single task), there is exactly one input module receiving task input, some number of side modules, and exactly one output module producing predictions. The output modules get trained using appropriate loss functions, with their gradients flowing backwards through the fully differentiable routing center into all modules.

Modules can operate in parallel as reads target the center vector from the previous time step. An unrolling of the multi-step process can be seen in Figure 1b. This figure illustrates the ability to arbitrarily route between modules between time steps This suggest a sequential nature of our model, even though application to static input is possible by allowing observing the input for multiple time steps.

We hypothesize that modules will use the center to route information through a chain of modules before producing the final output (see Section 4). For tasks that require producing an output at every time step, we repeat input frames to allow the model to process through multiple modules first, before producing an output. This is because communication between modules always spans a time step.[4]

## 2.2 Reading Mechanisms

We now discuss implementations of the reading mechanism $r^i(\Phi, \phi^i)$ and modules $f^i(c^i, x^i)$, as defined in Section 2.1. We draw a distinction between static and dynamic reading mechanisms for ThalNet. For static reading, $r^i(\Phi)$ is conditioned on independent parameters. For dynamic reading, $r^i(\Phi, \phi^i)$ is conditioned on the current corresponding module state, allowing the model to adapt its connectivity within a single sequence. We investigate the following reading mechanisms:

- **Linear Mapping.** In its simplest form, static reading consists of a fully connected layer $r(\Phi, \cdot) = W\Phi$ with weights $W \in \mathbb{R}^{|c| \times |\Phi|}$ as illustrated in Figure 2. This approach performs reasonably well, but can exhibit unstable learning dynamics and learns noisy weight matrices that are hard to interpret. Regularizing weights using L1 or L2 penalties does not help here since it can cause side modules to not get read from anymore.

- **Weight Normalization.** We found linear mappings with weight normalization [26] parameterization to be effective. For this, the context input is computed as $r(\Phi, \cdot) = \beta \frac{W}{|W|}\Phi$ with scaling factor $\beta \in \mathbb{R}$, weights $W \in \mathbb{R}^{|c| \times |\Phi|}$, and the Euclidean matrix norm $|W|$.

Normalization results in interpretable weights since increasing one weight pushes other, less important, weights closer to zero, as demonstrated in Section 4.

- **Fast Softmax.** To achieve dynamic routing, we condition the reading weight matrix on the current module features $\phi^i$. This can be seen as a form of fast weights, providing a biologically plausible method for attention [2, 27]. We then apply softmax normalization to the computed weights so that each element of the context is computed as a weighted average over center elements, rather than just a weighted sum. Specifically, $r(\Phi, \phi)_{(j)} = \left(e^{(W\phi+b)_{(j)}}/\sum_{k=1}^{|\Phi|} e^{(W\phi+b)_{(jk)}}\right)\Phi$ with weights $W \in \mathbb{R}^{|\phi| \times |\Phi| \times |c|}$, and biases $b \in \mathbb{R}^{|\Phi| \times |c|}$. While this allows for a different connectivity pattern at each time step, it introduces $|\phi^i + 1| \times |\Phi| \times |c^i|$ learned parameters per module.

- **Fast Gaussian.** As a compact parameterization for dynamic routing, we consider choosing each context element as a Gaussian weighted average of $\Phi$, with only mean and variance vectors learned conditioned on $\phi^i$. The context input is computed as $r(\Phi, \phi)_{(j)} = f\big((1, 2, \cdots, |\Phi|)|(W\phi + b)_{(j)}, (U\phi + d)_{(j)}\big)\Phi$ with weights $W, U \in \mathbb{R}^{|c| \times |\phi|}$, biases $b, d \in \mathbb{R}^{|c|}$, and the Gaussian density function $f(x|\mu, \sigma^2)$. The density is evaluated for each index in $\Phi$ based on its distance from the mean. This reading mechanism only requires $|\phi^i + 1| \times 2 \times |c^i|$ parameters per module and thus makes dynamic reading more practical.

Reading mechanisms could also select between modules on a high level, instead of individual feature elements. We do not explore this direction since it seems less biologically plausible. Moreover, we demonstrate that such knowledge about feature boundaries is not necessary, and hierarchical information flow emerges when using fine-grained routing (see Figure 4). Theoretically, this also allows our model to perform a wider class of computations.

## 3 Performance Comparison

We investigate the properties and performance of our model on several benchmark tasks. First, we compare reading mechanisms and module designs on a simple sequential task, to obtain a good configuration for the later experiments. Please refer to the supplementary material for the precise experiment description and results. We find that the weight normalized reading mechanism provides best performance and stability during training. We will use ThalNet models with four modules of configuration for all experiments in this section. To explore the performance of ThalNet, we now conduct experiments on three sequential tasks of increasing difficulty:

- **Sequential Permuted MNIST.** We use images from the MNIST [19] data set, the pixels of every image by a fixed random permutation, and show them to the model as a sequence of rows. The model outputs its prediction of the handwritten digit at the last time step, so that it must integrate and remember observed information from previous rows. This delayed prediction combined with the permutation of pixels makes the task harder than the static image classification task, with a multi-layer recurrent neural network achieving ~65 % test error. We use the standard split of 60,000 training images and 10,000 testing images.

- **Sequential CIFAR-10.** In a similar spirit, we use the CIFAR-10 [17] data set and feed images to the model row by row. We flatten the color channels of every row so that the model observes a vector of 96 elements at every time step. The classification is given after observing the last row of the image. This task is more difficult than the MNIST task, as the image show more complex and often ambiguous objects. The data set contains 50,000 training images and 10,000 testing images.

- **Text8 Language Modeling.** This text corpus consisting of the first $10^8$ bytes of the English Wikipedia is commonly used as a language modeling benchmark for sequential models. At every time step, the model observes one byte, usually corresponding to 1 character, encoded as a one-hot vector of length 256. The task it to predict the distribution of the next character in the sequence. Performance is measured in bits per character (BPC) computed as $-\frac{1}{N}\sum_{i=1}^{N}\log_2 p(x_i)$. Following Cooijmans et al. [4], we train on the first 90% and evaluate performance on the following 5% of the corpus.

For the two image classification tasks, we compare variations of our model to a stacked Gated Recurrent Unit (GRU) [3] network of 4 layers as baseline. The variations we compare are different

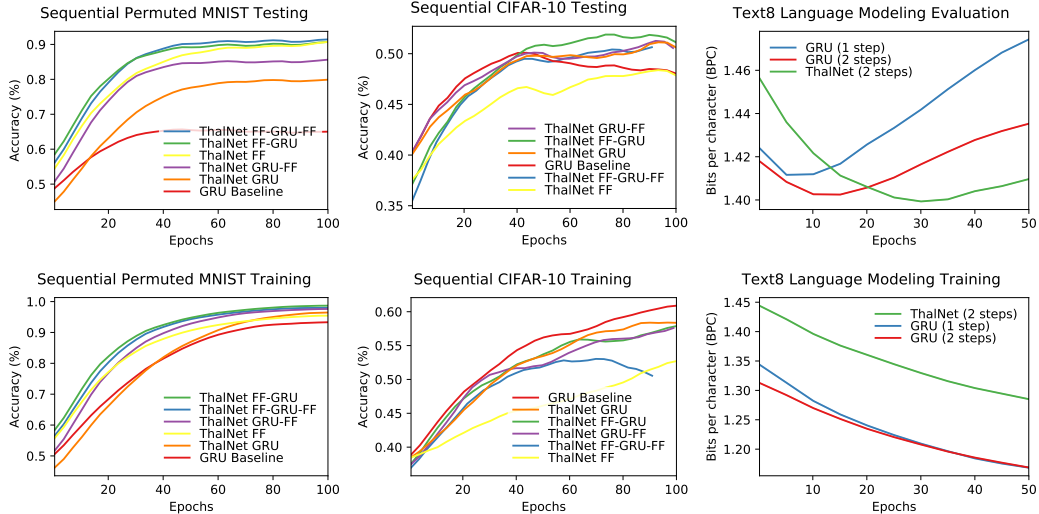

Figure 3: Performance on the permuted sequential MNIST, sequential CIFAR, and text8 language modeling tasks. The stacked GRU baseline reaches higher training accuracy on CIFAR, but fails to generalize well. On both tasks, ThalNet clearly outperforms the baseline in testing accuracy. On CIFAR, we see how recurrency within the modules speeds up training. The same pattern is shows for the text8 experiment, where ThalNet using 12M parameters matches the performance of the baseline with 14M parameters. The step number 1 or 2 refers to repeated inputs as discussed in Section 2. We had to smooth the graphs using a running average since the models were evaluated on testing batches on a rolling basis.

choices of feed-forward layers and GRU layers for implementing the modules $f^i(c^i, x^i)$: We test with two fully connected layers (FF), a GRU layer (GRU), fully connected followed by GRU (FF-GRU), GRU followed by fully connected (GRU-FF), and a GRU sandwiched between fully connected layers (FF-GRU-FF).[5] For all models, we pick the largest layer sizes such that the number of parameters does not exceed 50,000. Training is performed for 100 epochs on batches of size 50 using RMSProp [30] with a learning rate of $10^{-3}$.

For language modeling, we simulate ThalNet for 2 steps per token, as described in Section 2 to allow the output module to read information about the current input before making its prediction. Note that on this task, our model uses only half of its capacity directly, since its side modules can only integrate longer-term dependencies from previous time steps. We run the baseline once without extra steps and once with 2 steps per token, allowing it to apply its full capacity once and twice on each token, respectively. This makes the comparison a bit difficult, but only by favouring the baseline. This suggests that architectural modifications, such as explicit skip-connections between modules, could further improve performance.

The Text8 task requires larger models. We train ThalNet with 4 modules of a size 400 feed forward layer and a size 600 GRU layer each, totaling in 12 million model parameters. We compare to a standard baseline in language modeling, a single GRU with 2000 units, totaling in 14 million parameters. We train on batches of 100 sequences, each containing 200 bytes, using the Adam optimizer [15] with a default learning rate of $10^{-3}$. We scale down gradients exceeding a norm of 1. Results for 50 epochs of training are shown in Figure 3. The training took about 8 days for ThalNet with 2 steps per token, 6 days for the baseline with 2 steps per token, and 3 days for the baseline without extra steps.

Figure 3 shows the training and testing and training curves for the three tasks described in this section. ThalNet outperforms standard GRU networks in all three tasks. Interestingly, ThalNet experiences a

much smaller gap between training and testing performance than our baseline – a trend we observed across all experimental results.

On the Text8 task, ThalNet scores 1.39 BPC using 12M parameters, while our GRU baseline scores 1.41 BPC using 14M parameters (lower is better). Our model thus slightly improves on the baseline while using fewer parameters. This result places ThalNet in between the baseline and regularization methods designed for language modeling, which can also be applied to our model. The baseline performance is consistent with published results of LSTMs with similar number of parameters [18].

We hypothesize the information bottleneck at the reading mechanism acting as an implicit regularizer that encourages generalization. Compared to using one large RNN that has a lot of freedom of modeling the input-output mapping, ThalNet imposes local structure to how the input-output mapping can be implemented. In particular, it encourages the model to decompose into several modules that have stronger intra-connectivity than extra-connectivity. Thus, to some extend every module needs to learn a self-contained computation.

## 4  Hierarchical Connectivity Patterns

Using its routing center, our model is able to learn its structure as part of learning to solve the task. In this section, we explore the emergent connectivity patterns. We show that our model learns to route features in hierarchical ways as hypothesized, including skip connections and feedback connections. For this purpose, we choose the text8 corpus, a medium-scale language modeling benchmark consisting of the first $10^8$ bytes of Wikipedia, preprocessed for the Hutter Prize [21]. The model observes one one-hot encoded byte per time step, and is trained to predict its future input at the next time step.

We use comparably small models to be able to run experiments quickly, comparing ThalNet models of 4 FF-GRU-FF modules with layer sizes 50, 100, 50 and 50, 200, 50. Both experiments use weight normalized reading. Our focus here is on exploring learned connectivity patterns. We show competitive results on the task using larger models in Section 3.

We simulate two sub time steps to allow for the output module to receive information of the current input frame as discussed in Section 2. Models are trained for 50 epochs on batches of size 10 containing sequences of length 50 using RMSProp with a learning rate of $10^{-3}$. In general, we observe different random seeds converging to similar connectivity patterns with recurring elements.

### 4.1  Trained Reading Weights

Figure 4 shows trained reading weights for various reading mechanisms, along with their connectivity graphs that were manually deduced.[6] Each image represents a reading weight matrix for the modules 1 to 4 (top to bottom). Each pixel row shows the weight factors that get multiplied with $\Phi$ to produce a single element of the context vector of that module. The weight matrices thus has dimensions of $|\Phi| \times |c^i|$. White pixels represent large magnitudes, suggesting focus on features at those positions.

The weight matrices of weight normalized reading clearly resemble the boundaries of the four concatenated module features $\phi^1, \cdots, \phi^4$ in the center vector $\Phi$, even though the model has no notion of the origin and ordering of elements in the center vector.

A similar structure emerges with fast softmax reading. These weight matrices are sparser than the weights from weight normalization. Over the course of a sequence, we observe some weights staying constant while others change their magnitudes at each time step. This suggests that optimal connectivity might include both static and dynamic elements. However, this reading mechanism leads to less stable training. This problem could potentially alleviated by normalizing the fast weight matrix.

With fast Gaussian reading, we see that the distributions occasionally tighten on specific features in the first and last modules, the modules that receive input and emit output. The other modules learn large variance parameters, effectively spanning all center features. This could potentially be addressed by reading using mixtures of Gaussians for each context element instead. We generally find that weight normalized and fast softmax reading select features with in a more targeted way.

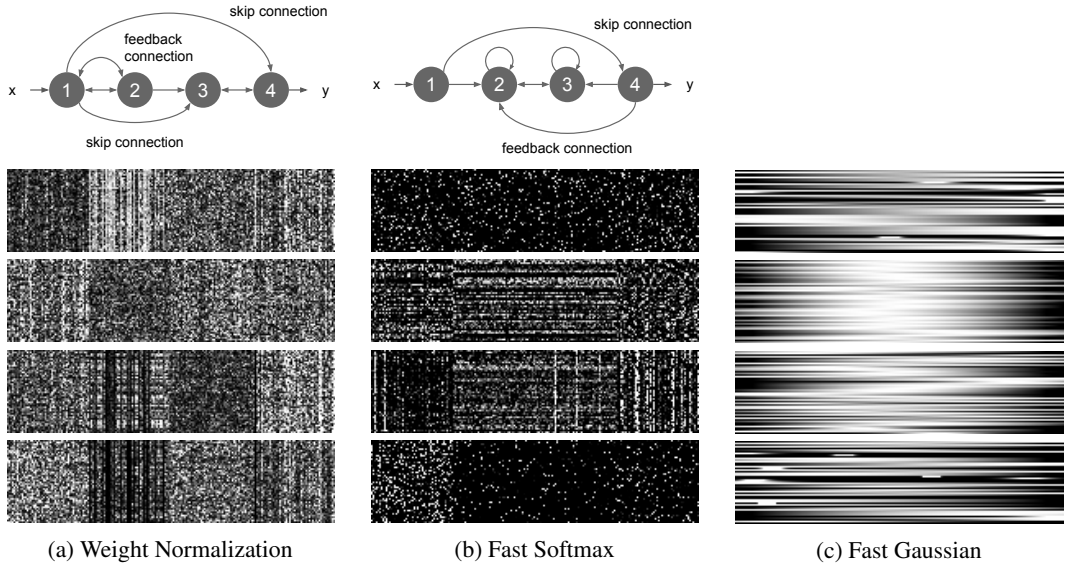

(a) Weight Normalization          (b) Fast Softmax          (c) Fast Gaussian

Figure 4: Reading weights learned by different reading mechanisms with 4 modules on the text8 language modeling task, alongside manually deducted connectivity graphs. We plot the weight matrices that produce the context inputs to the four modules, top to bottom. The top images show focus of the input modules, followed by side modules, and output modules at the bottom. Each pixel row gets multiplied with the center vector $\Phi$ to produce one scalar element of the context input $c^i$. We visualize the magnitude of weights between the 5 % to the 95 % percentile. We do not include the connectivity graph for Fast Gaussian reading as its reading weights are not clearly structured.

## 4.2 Commonly Learned Structures

The top row in Figure 4 shows manually deducted connectivity graphs between modules. Arrows represent the main direction of information flow in the model. For example, the two incoming arrows to module 4 in Figure 4a indicate that module 4 mainly attends to features produced by modules 1 and 3. We infer the connections from the larger weight magnitudes in the first and third quarters of the reading weights for module 4 (bottom row).

A typical pattern that emerges during the experiments can be seen in the connectivity graphs of both weight normalized and fast softmax reading (Figures 4a and 4b). Namely, the output module reads features directly from the input module. This direction connection is established early on during training, likely because this is the most direct gradient path from output to input. Later on, the side modules develop useful features to support the input and output modules.

In another pattern, one module reads from all other modules and combines their information. In Figure 4b, module 2 takes this role, reading from modules 1, 3, 4, and distributing these features via the input module. In additional experiments with more than four modules, we observed this pattern to emerge predominantly. This connection pattern provides a more efficient way of information sharing than cross-connecting all modules.

Both connectivity graphs in Figure 4 include hierarchical computation paths through the modules. They include learn skip connections, which are known to improve gradient flow from popular models such as ResNet [12], Highway networks [29], and DenseNet [14]. Furthermore, the connectivity graphs contain backward connections, creating feedback loops over two or more modules. Feedback connections are known to play a critical role in the neocortex, which inspired our work [7].

## 5 Related Work

We describe a recurrent mixture of experts model, that learns to dynamically pass information between the modules. Related approaches can be found in various recurrent and multi-task methods as outlined in this section.

**Modular Neural Networks.** ThalNet consists of several recurrent modules that interact and exploit each other. Modularity is a common property of existing neural models. [5] learn a matrix of tasks and robot bodies to improve both multitask and transfer learning. [1] learn modules modules specific to objects present in the scene, which are selected by an object classifier. These approaches specify modules corresponding to a specific task or variable manually. In contrast, our model automatically discovers and exploits the inherent modularity of the task and does not require a one-to-one correspondence of modules to task variables.

The Column Bundle model [23] consists of a central column and several mini-columns around it. While not applied to temporal data, we observe a structural similarity between our modules and the mini-columns, in the case where weights are shared among layers of the mini-columns, which the authors mention as a possibility.

**Learned Computation Paths.** We learn the connectivity between modules alongside the task. There are various methods in the multi-task context that also connectivity between modules. Fernando et al. [6] learn paths through multiple layers of experts using an evolutionary approach. Rusu et al. [25] learn adapter connections to connect to fixed previously trained experts and exploit their information. These approaches focus on feed-forward architectures. The recurrency in our approach allows for complex and flexible computational paths. Moreover, we learn interpretable weight matrices that can be examined directly without performing costly sensitivity analysis.

The Neural Programmer Interpreted presented by Reed and De Freitas [24] is related to our dynamic gating mechanisms. In their work, a network recursively calls itself in a parameterized way to perform tree-shaped computations. In comparison, our model allows for parallel computation between modules and for unrestricted connectivity patterns between modules.

**Memory Augmented RNNs.** The center vector in our model can be interpreted as an external memory, with multiple recurrent controllers operating on it. Preceding work proposes recurrent neural networks operating on external memory structures. The Neural Turing Machine proposed by Graves et al. [9], and follow-up work [10], investigate differentiable ways to address a memory for reading and writing. In the ThalNet model, we use multiple recurrent controllers accessing the center vector. Moreover, our center vector is recomputed at each time step, and thus should not be confused with a persistent memory as is typical for model with external memory.

# 6 Conclusion

We presented ThalNet, a recurrent modular framework that learns to pass information between neural modules in a hierarchical way. Experiments on sequential and permuted variants of MNIST and CIFAR-10 are a promising sign of the viability of this approach. In these experiments, ThalNet learns novel connectivity patterns that include hierarchical paths, skip connections, and feedback connections.

In our current implementation, we assume the center features to be a vector. Introducing a matrix shape for the center features would open up ways to integrate convolutional modules and similarity-based attention mechanisms for reading from the center. While matrix shaped features are easily interpretable for visual input, it is less clear how this structure will be leveraged for other modalities.

A further direction of future work is to apply our paradigm to tasks with multiple modalities for inputs and outputs. It seems natural to either have a separate input module for each modality, or to have multiple output modules that can all share information through the center. We believe this could be used to hint specialization into specific patterns and create more controllable connectivity patterns between modules. Similarly, we an interesting direction is to explore the proposed model can be leveraged to learn and remember a sequence of tasks.

We believe modular computation in neural networks will become more important as researchers approach more complex tasks and employ deep learning to rich, multi-modal domains. Our work provides a step in the direction of automatically organizing neural modules that leverage each other in order to solve a wide range of tasks in a complex world.

## Footnotes

*Work done during an internship with Google Brain.

[2]In practice, we experiment with both feed forward and recurrent implementations of the modules $f^i$. For simplicity, we omit the hidden state used in recurrent modules in our notation.

[3]The reading mechanism is conditioned on both $\Phi_t$ and $\phi_t^i$ separately as the merging does not preserve $\phi_t^i$ in the general case.

[4]Please refer to Graves [8] for a study of a similar approach.

[5]Note that the modules require some amount of local structure to allow them to specialize. Implementing the modules as a single fully connected layer recovers a standard recurrent neural network with one large layer.

[6]Developing formal measurements for this deduction process seems beneficial in the future.

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
