[Supplementary Material]

# Supplementary Material for Learning Hierarchical Information Flow with Recurrent Neural Modules

## A  Module Designs and Reading Mechanisms

(a) Module designs
(b) Reading mechanisms

Figure 5: Test performance on the Sequential MNIST task grouped by module design (left) and reading mechanism (right). Plots show the top, median, and bottom accuracy over the other design choices. Recurrent modules train faster than pure fully connected modules and weight normalized reading is both stable and performs best. FF-GRU-FF modules perform similarly to FF-GRU while limiting the size of the center.

We use a sequential variant of MNIST [21] to compare the reading mechanisms described in Section 2.2, along with implementations of the module function. In Sequential MNIST, the model observes handwritten digits of $28 \times 28$ pixels from top to bottom, one row per time step. The prediction is given at the last time step, so that the model has to integrate and remember observed information over the sequence. This makes the task more challenging than in the static setting with a multi-layer recurrent network achieving ~7 % error on this task.

To implement the modules $f^i(c^i, x^i)$ we test various combinations of fully connected and recurrent layers of Gated Recurrent Units (GRU) [3]. Modules require some amount of local structure to allow them to specialize.[7] We test with two fully connected layers (FF), a GRU layer (GRU), fully connected followed by GRU (FF-GRU), GRU followed by fully connected (GRU-FF), and a GRU sandwiched between fully connected layers (FF-GRU-FF). In addition, we compare performance to a stacked GRU baseline with 4 layers. For all models, we pick the largest layer sizes such that the number of parameters does not exceed 50,000.

We train for 100 epochs on batches of size 50 using RMSProp [33] with a learning rate of $10^{-3}$. Figure 5 shows the test accuracy of module designs and reading mechanisms. ThalNet outperforms the stacked GRU baseline in most configurations. We assume that the structure imposed by our model acts as a regularizer. We perform a further performance comparison in Section 3.

Results for module designs are shown in Figure 5a in the appendix. We observe a benefit of recurrent modules as they exhibit faster and more stable training than fully connected modules. This could be explained by the fact that pure fully connected modules have to learn to use the routing center to store information over time, which is a long feedback loop. Having a fully connected layer before the recurrent layer also significantly improves performance. A fully connected layer after the GRU let us produce compact feature vectors $\phi^i$ that scale better to large modules, although we find FF-GRU to be beneficial in later experiments (Section 3).

Results for the reading mechanisms area shown in Figure 5b. The reading mechanism only has a small impact on the model performance. We find weight normalized reading to yield more stable performance than linear or fast softmax reading. For all further experiments, we use weight normalized reading due to both its stability and predictive performance. We do not include results for fast Gaussian reading here, as it performed below the performance range of the other methods.

## B  Interpretation as Recurrent Mixture of Experts

ThalNet can route information from the input to the output over multiple time steps. This enables it to trade off shallow and deep computation paths. To understand this, we view ThalNet as a smooth mixture of experts model [16], where the modules $F = (f^1, \cdots, f^I)$ are the recurrent experts. Each module outputs its features to the center vector $\Phi_t$. A linear combination of $\Phi_t$ is read at the next time step, which effectively performs a mixing of expert outputs. Compared to the recurrent mixture of experts model presented by Shazeer et al. [30], our model can recurrently route information through the mixture of multiple times, increasing the number of mixture compounds.

To highlight two extreme cases, the modules could read from identical locations in the center. In this case, the model does a wide and shallow computation over 1 time step, analogous to Graves [8]. In the other extreme, each module reads from a different module, recovering a hierarchy of recurrent layers. This gives a deep but narrow computation stretched over multiple time steps. In between, there exist a spectrum of complex patterns of information flow with differing and dynamic computation depths. This is comparable to DenseNet [15], which also blends information from paths of different computational depth, although in a purely feed-forward model.

Using state-less modules, our model could still leverage the recurrence between the modules and the center to store information over time. However, this bounds the number of distinct computation steps that ThalNet could apply to an input. Using recurrent modules, the computation steps can change over time, increasing the flexibility of the model. Recurrent modules give a stronger prior for using feedback and shows improved performance in our experiments.

## C  Comparison to Long Short-Term Memory

When viewing the Equations 1 – 4 in the model definition (Section 2), one might think how our model compares to Long Short-Term Memory (LSTM) [14]. However, there exists only a limited similarity between the two models. Empirically, we observed that LSTMs performed similarly to our GRU baselines when given the same parameter budget.

LSTM's context vector $c_t$ is processed element-wise, while ThalNet's routing center cross-connects modules. LSTM's hidden output $h_t$ is a better candidate for comparison with ThalNet's center features $\Phi$, which allows us to relate the recurrent weight matrix of an LSTM layer to the linear version of our reading mechanism.

We could relate each ThalNet module to a set of multiple LSTM units. However, LSTM units perform separate scalar computations, while our modules can learn complex interactions between multiple features at each time step. Alternatively, we could see LSTM units as very small ThalNet modules, reading exactly four context elements each, namely for the input and the three gates. However, the computational capacity and local structure of individual LSTM units is not comparable to that of the ThalNet modules used in our work.

## Footnotes

[7]Implementing the modules as a single fully connected layer recovers a standard recurrent neural network with one large layer.