[Reviews · NeurIPS 2017]

Reviewer 1



I think overall the idea maybe nice but it is rather unclear why this should work. I think the paper is rather preliminary both in understanding its own idea and putting it into literature and math context. Moreover the experiments show that something is learned but not whether it is the thing promised. There are some interesting similarities to hmms mrfs and mixture of experts. All of them have at least some convergence proofs etc. the new idea not. It is unclear whether ultimately the right structure or one closeby or very far off is inferred... In that sense the experiments are only a first step to convincing me.

Reviewer 2



This paper introduces a neural architecture for sequential tasks where connections between stacked modules are learned during the model training. The approach presented can contribute to the import goal of learning model architectures from the data. The examples presented show convincingly that indeed non-trivial connectivity patterns can be learned.

Reviewer 3



The rebuttal has addressed my concerns very well. I would upgrade my score from 6 to 7. This paper proposes a new gated recurrent model, which employs a routing center to summarize the information from multi modules around it in each time step. The model is expected to learn to route information flow hierarchically. The paper is not written very clearly. The idea is novel. Pros: 1. The idea inspired by the thalamus is novel. Cons: 1. The experiments on image datasets (MNIST and CIFAR) are not not convincing. Why not try the real-world sequence (time series) data set? 2. How about the comparison with the LSTM? LSTM also has a cell which somewhat serves like a routing center between different time steps. 3. In the experiment on MNIST (section 3.1), how many modules are used in the model in each time step? It seems there is only one module and cannot demonstrate the effectiveness of routing center.

Reviewer 4



The paper introduces a modular architecture that has a central routing module and other in/out modules, akin to the role of thalamus/neocortex regions in the brain. Initial experiments show promises. I like the idea of the paper, which was nicely written. Some module details are missing (the read function r, merging function m and the module function f), possibly because the space limitation, but it would be very useful to have in a supplement. The main cons are (i) the experiments are not super-strong; (ii) the search for the best module/routing functions is quite preliminary. However I believe that the direction is fruitful and worth exploring further. In related work, there is a contemporary architecture with similar intention (although the specific targets were different, i.e., multi-X learning, where X = instance, view, task, label): "On Size Fit Many: Column Bundle for Multi-X Learning", Trang Pham, Truyen Tran, Svetha Venkatesh. arXiv preprint arXiv: 1702.07021.